# RIPK1-RIPK3-MLKL-dependent necrosis promotes the aging of mouse male reproductive system

Dianrong Li, Lingjun Meng, Tao Xu, Yaning Su, Xiao Liu, Zhiyuan Zhang, Xiaodong Wang*

National Institute of Biological Sciences, Beijing, China

**Abstract** A pair of kinases, RIPK1 and RIPK3, as well as the RIPK3 substrate MLKL cause a form of programmed necrotic cell death in mammals termed necroptosis. We report here that male reproductive organs of both *Ripk3-* and *Mlkl-*knockout mice retain 'youthful' morphology and function into advanced age, while those of age-matched wild-type mice deteriorate. The RIPK3 phosphorylation of MLKL, the activation marker of necroptosis, is detected in spermatogonial stem cells in the testes of old but not in young wild-type mice. When the testes of young wild-type mice are given a local necroptotic stimulus, their reproductive organs showed accelerated aging. Feeding of wild-type mice with an RIPK1 inhibitor prior to the normal onset of age-related changes in their reproductive organs blocked the appearance of signs of aging. Thus, necroptosis in testes promotes the aging-associated deterioration of the male reproductive system in mice.

## Introduction

Necroptosis is a form of programmed necrotic cell death caused by the tumor necrosis factor family of cytokines (*Christofferson and Yuan, 2010*; *Vandenabeele et al., 2010*). In response to the activation of TNF receptor family members, receptor-interacting kinase 1 (RIPK1) is recruited to the cytosolic side of the receptor and its kinase activity is activated (*Holler et al., 2000*). RIPK1 then interacts with and phosphorylates a related kinase, RIPK3, leading to its activation (*Cho et al., 2009*; *Degterev et al., 2008*; *He et al., 2009*; *Zhang et al., 2009*). If the cells also happen to have their caspase-8 activity inhibited, either through interaction with its cellular inhibitor cFLIP or through the action of viral or chemical inhibitors, RIPK3 drives the cell fate towards necroptosis (*He et al., 2009*; *Holler et al., 2000*). Necroptosis can be inhibited by RIPK1 kinase inhibitor compounds, and can be promoted by small molecule Smac mimetics, which shifts RIPK1 function from NF-κB activation to activation of RIPK3 (*Degterev et al., 2008*; *Wang et al., 2008*). Once active, RIPK3 then phosphorylates a pseudokinase called mixed lineage kinase domain-like protein (MLKL) (*Sun et al., 2012*). MLKL normally exists as an inactive monomer in the cytosol. Upon RIPK3 phosphorylation on serine 357 and threonine 358 of human MLKL or the mouse equivalent of serine 345, serine 347, and threonine 349, MLKL forms oligomers and translocates to the plasma membrane, where it disrupts membrane integrity, resulting in necrotic cell death (*Cai et al., 2014*; *Chen et al., 2014*; *Hildebrand et al., 2014*; *Murphy et al., 2013*; *Rodriguez et al., 2016*; *Sun et al., 2012*; *Wang et al., 2014*).

Necroptosis is known to have important functions under pathological conditions of microbial infections or tissue damage since *Ripk3* knockout mice show defects in defending microbial infections or manifest less tissue damage in a variety of chemical or ischemic reperfusion induced tissue damage models (*Cho et al., 2009*; *He et al., 2009*; *Robinson et al., 2012*; *Upton et al., 2010*; *Zhou and Yuan, 2014*). However, mice with *Ripk3* or *Mlkl* gene knockout are remarkably normal

*For correspondence:
wangxiaodong@nibs.ac.cn

**Competing interests:** The authors declare that no competing interests exist.

without any noticeable developmental, physiological, or fertility defects (*Murphy et al., 2013*; *Newton et al., 2004*; *Wu et al., 2013*). Therefore, under what physiological conditions necroptosis happens in which tissue remains the biggest unanswered question in this research field.

While conducting a study investigating the impact of necroptosis on the progression of atherosclerosis (*Meng et al., 2015*), we serendipitously found that the male reproductive organ of mice with *Ripk*3 and *Mlkl* gene knockout looked remarkably young even at advanced ages. A comprehensive study present here revealed that necroptosis functions in promoting the aging of male reproductive system in mice.

## Results

### The aging of reproductive organs is delayed in *Ripk*3-knockout mice

We first noticed that the 18-month-old *Ripk*3-knockout (*Ripk*3$^{-/-}$) mice of the C57BL/6 strain looked thinner than the age-matched wild-type (WT, *Ripk*3$^{+/+}$) mice of the same strain that were housed under the same conditions (*Figure 1A*). The average weight of 18-month-old wild-type mice was 46 grams, significantly more than of 37 grams of weight of the age-matched *Ripk*3-knockout mice (*Figure 1B*). The weights of 4-month-old wild-type and *Ripk*3-knockout mice, on the other hand, were indistinguishable (*Figure 1B*). In addition to differences in whole body weights, the seminal vesicles, an auxiliary gland in the mouse male reproductive system, appeared to be quite different between 18-month-old *Ripk*3-knockout and wild-type mice (*Figure 1C*). The weights of the seminal vesicles from 18-month-old wild-type mice (n = 33) ranged from ~1,000 mg to 4,500 mg, while the weights of the same organ from the age-matched *Ripk*3-knockout mice (n = 30) were mostly below 1,000 mg (*Figure 1D*).

It is known that seminal vesicles become enlarged as mice get old, presumably due to secondary growth in response to declining of testis functions that is similar to prostate hypertrophy in human (*Finch and Girgis, 1974*; *Pettan-Brewer and Treuting, 2011*). The difference in seminal vesicles from wild-type and *Ripk*3-knockout mice become noticeable after one year of life, and become increasingly evident over time. The seminal vesicles from wild-type mice continue to grow, whereas the seminal vesicles from the *Ripk*3-knockout mice did not change in size from 4 months to 24 months (*Figure 1—figure supplement 1A and B*). There were no obvious differences in the overall anatomical structure of seminal vesicles between wild-type and *Ripk*3-knockout mice (*Figure 1C* and *Figure 1—figure supplement 1A*). Close examination revealed that the epithelium of the seminal vesicles from 18-month-old wild-type mice showed irregularities, with spaces separating the epithelium and the liquid compartment, whereas the seminal-vesicle epithelial cells from the age-matched *Ripk*3-knockout mice were tightly packed, just as they are in young mice (*Figure 1—figure supplement 1C*)

The seminal vesicles of mice are anatomically simple, consisting of only an epithelial layer that envelopes a liquid compartment (*Gonzales, 2001*). Therefore, the difference in seminal vesicles between wild-type and *Ripk*3-knockout mice did not offer much mechanistic insight what caused such a phenotype. We further studied the testes of wild-type and *Ripk*3-knockout mice. By the time mice reached 18 months of age, the wild-type testes started to appear atrophic, and weighed less than *Ripk*3-knockout testes (*Figure 1—figure supplement 1D and E*). Consistently, the testosterone level showed a dramatic drop as wild-type mice aged from 4 to 18 months, whereas the testosterone levels hardly decreased at all in *Ripk*3-knockout mice over the same period (*Figure 1E*). Moreover, the typical age-related increase in sex hormone-binding globulin (SHBG) (*Vermeulen et al., 1996*) that is known to occur in wild-type mice was not observed in *Ripk*3-knockout mice (*Figure 1—figure supplement 2A*). Interestingly, the levels of two endocrine factors secreted by the pituitary gland, LH and FSH (*Cooke and Saunders, 2002*), did not differ between wild-type and *Ripk*3-knockout mice; both dropped significantly as mice aged from 4 months to 18 months (*Figure 1—figure supplement 2B and C*). This finding indicated that the difference in aging of reproductive system between old wild-type and *Ripk*3-knockout mice may result from local changes in testis.

Unlike what often happens in human upon reproductive organ aging, we did not notice any apparent anatomical difference in the anterior, dorsal, ventral, or lateral prostate by an hematoxylin and eosin (H and E) staining of mouse prostates (*Pettan-Brewer and Treuting, 2011*) sections of

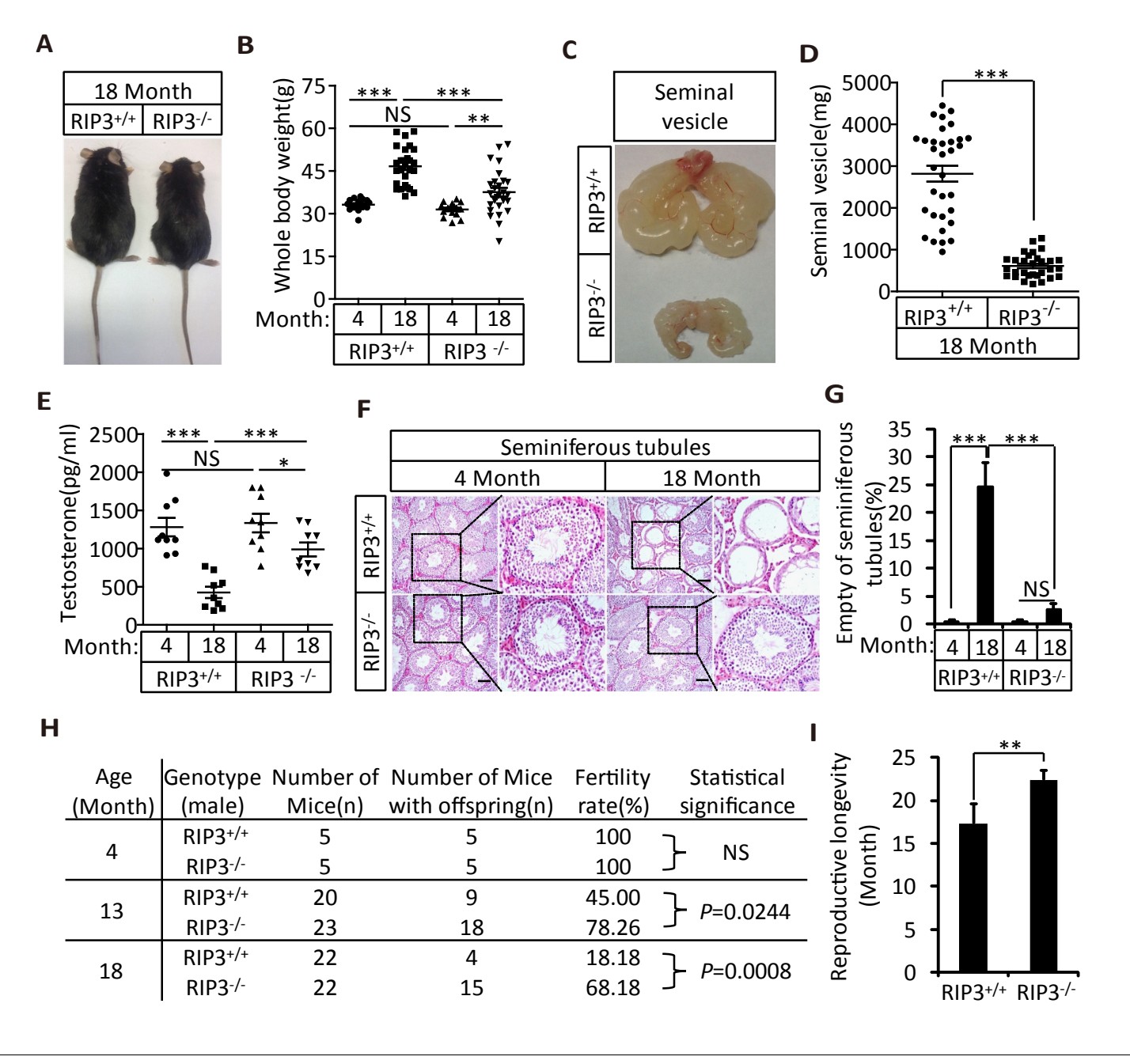

**Figure 1.** *Ripk3⁻ᐟ⁻* mice maintained their reproductive system function at an advanced age. (**A, B**) Macroscopic features and weights of *Ripk3⁺ᐟ⁺* (wild-type) and *Ripk3⁻ᐟ⁻* (*Ripk3*-knockout) male mice. *Ripk3⁺ᐟ⁺* (4 Month, n = 16; 18 Month, n = 27) and *Ripk3⁻ᐟ⁻*(4 Month, n = 16; 18 Month, n = 27) male mice were photographed and weighed. Data represent the mean ±the standard error of the mean (s.e.m). **p<0.01, ***p<0.001. p values were determined with unpaired Student's *t*-tests. NS, not significant. (**C, D**) Macroscopic features and weights of seminal vesicles. Mice were sacrificed at 18 months of age, and the seminal vesicles from *Ripk3⁺ᐟ⁺* (n = 33) and *Ripk3⁻ᐟ⁻* (n = 30) mice were photographed and weighed. Data represent the mean ±s.e.m. ***p<0.001. p values were determined with unpaired Student's *t*-tests. (**E**) Serum testosterone levels of mice assayed using ELISA. Mice were sacrificed, and the testosterone levels in serum from *Ripk3⁺ᐟ⁺* (4 Month, n = 9; 18 Month, n = 9) and *Ripk3⁻ᐟ⁻* (4 Month, n = 9; 18 Month, n = 9) mice were measured using an ELISA kit for testosterone. Data represent the mean ±s.e.m. *p<0.05, ***p<0.001. p values were determined with unpaired Student's *t*-tests. NS, not significant. (**F, G**) H and E of testis sections from *Ripk3⁺ᐟ⁺* and *Ripk3⁻ᐟ⁻* mice. *Ripk3⁺ᐟ⁺* (4 Months, n = 10; 18 Months, n = 10) and *Ripk3⁻ᐟ⁻* (4 Months, n = 10; 18 Months, n = 10) mice were sacrificed and testes were harvested and stained with H and E in (**F**). The number of empty seminiferous tubules was counted based on H and E staining and quantification in (**G**), empty seminiferous tubules were counted in five fields per testis. Scale bar, 100 μm. Data represent the mean ± S.D. ***p<0.001. p values were determined with unpaired Student's *t*-tests. NS, not significant. (**H**) Summary of the fertility rates of *Ripk3⁺ᐟ⁺* and *Ripk3⁻ᐟ⁻* mice. One male mice of a given age was mated with a pairs of 10-week-old wild-type female mice
*Figure 1 continued on next page*

*Figure 1 continued*

for 3 months; females were replaced every 2 weeks. The number of male mice with reproductive capacity was counted (see Materials and methods). p values were determined using chi-square tests. (I) Reproductive longevity. When $Ripk3^{+/+}$ (n = 12) and $Ripk3^{-/-}$ (n = 12) male mice were 2 months old, they were continuously mated with a pairs of 10-week-old female mice until pregnancies ceased; females were replaced every 2 months. The ages of the males at which their last litter was sired was recorded (calculated as the age at birth of the litter less 21 days, see Materials and methods). Data represent the mean ± S.D. **p<0.01. p values were determined with unpaired Student's *t*-tests.

The following source data and figure supplements are available for figure 1:

**Source data 1.** Summary of the fertility rates and mortality rates of the offspring of 4- or 18-month-old $Ripk3^{+/+}$ and $Ripk3^{-/-}$ male mice.

**Source data 2.** Summary of the fertility rates and mortality rates of the offspring of 13-month-old $Ripk3^{+/+}$ and $Ripk3^{-/-}$ male mice.

**Figure supplement 1.** Morphological changes in seminal vesicles and testis during aging.

**Figure supplement 2.** The levels of the pituitary endocrine hormones LH and FSH decline normally in $Ripk3^{+/+}$ and $Ripk3^{-/-}$ mice.

**Figure supplement 3.** No morphological differences were apparent in the prostates of $Ripk3^{+/+}$ and $Ripk3^{-/-}$ mice.

**Figure supplement 4.** Morphological changes in seminiferous tubules in 36-month-old mice.

**Figure supplement 5.** $Ripk3^{-/-}$ mice have higher sperm counts than $Ripk3^{+/+}$ mice at an advanced age.

**Figure supplement 6.** Histological analysis of various organs of $Ripk3^{+/+}$ and $Ripk3^{-/-}$ mice.

**Figure supplement 7.** Increase of rates of birth defects and oxidative damage in sperm from aged $Ripk3^{-/-}$ mice.

**Figure supplement 8.** Weights of fat and seminal vesicles from 18-month-old $Ripk3^{+/+}$ and $Ripk3^{-/-}$ male mice.

young (4 month) or old (18 month) mice of either the wild-type or *Ripk*3-knockout genotypes (*Figure 1—figure supplement 3*).

## Knockout of *Ripk*3 prevents the depletion of cells in the seminiferous tubules in aged testes

As a male mouse becomes sexually mature, the central lumens of seminiferous tubules in its testes begin to fill with sperm generated from the surrounding spermatogonial stem cells. The spermatogonial stem cells and spermatocytes are supported by Sertoli cells, which provide trophic factors and structural support for spermatogenesis (*Cooke and Saunders, 2002*). Sperm are then transferred and stored in the epididymis, from where mature sperm are ejected. After mixing with fluids from the seminal vesicles and prostate, the sperm travel alone the ejaculation track, where semen is formed (*Cooke and Saunders, 2002*).

When testes from 4-month-old and 18-month-old wild-type and *Ripk*3-knockout mice were dissected and their cross sections were examined under a microscope, cells in many of the seminiferous tubules from the 18-month-old wild-type mice were lost, given the seminiferous tubules an 'empty' appearance (*Figure 1F and G*). In contrast, the central lumens of the seminiferous tubules of 4-month-old wild-type and *Ripk*3-knockout mice are fully surrounded with cells, and are filled with sperm. Strikingly, the seminiferous tubules of 18-month-old *Ripk*3-knockout mice looked no different than those of 4-month-old mice (*Figure 1F and G*). Even more dramatically, when testis sections from 36-month-old mice were examined, close to half of seminiferous tubules of wild-type mice were empty, while more than 90% of those from the *Ripk*3-knockout mice still appeared normal (*Figure 1—figure supplement 4*).

Sperm from the seminiferous tubules travel to the epididymis, where they mature and are stored prior to ejaculation (*Cooke and Saunders, 2002*). Similar to the phenotypes observed in the seminiferous tubules, most of the epididymides from 18-month-old wild-type mice had few sperm, whereas most of the epididymides of age-matched *Ripk*3-knockout mice were full of sperm (*Figure 1—figure*

*supplement 5A*). The sperm counts in epididymes increased steadily during development and peaked at four months of age, and there was little difference in the sperm counts between wild-type and *Ripk*3-knockout mice up to this time (*Figure 1—figure supplement 5B*). The sperm counts of wild-type mice then started to decline, while those of *Ripk*3-knockout mice remained steady until 12 months of age. Even at 24 months, the sperm counts of *Ripk*3-knockout mice were still comparable with those of 4-month-old wild-type mice (*Figure 1—figure supplement 5B*).

## Knockout of R*ipk*3 prevents age-associated decline of reproductive capacity

To test if the morphologically 'young' reproductive system from aged *Ripk*3-knockout mice remain functional, we set up breeding experiments that mated 4-month-old, 13-month-old, and 18-month-old male mice with pairs of 10-week-old wild-type female mice. As summarized in *Figure 1H*, both wild-type and *Ripk*3-knockout 4-month-old male mice were fully fertile, and both groups sired a similar number of pups (*Figure 1—source data 1*). However, for 13-month-old mice, only 9 of the 20 (45%)wild-type male mice sired pups, while 18 out of 23 (78%)*Ripk*3-knockout males remained fertile (*Figure 1H* and *Figure 1—source data 1* and *2*). The difference was even more obvious with the 18-month-old male mice. Only 4 out of 22 (18%)wild-type male mice were still fertile at this age, whereas 15 out of 22 (68%)*Ripk*3-knockout male mice remained fertile (*Figure 1H* and *Figure 1—source data 1*). We subsequently measured the reproductive longevity of wild-type and *Ripk*3-knockout male mice by pairing a pair of 10-weeks-old female mice with each male in a cage and switch out a fresh pair of females every other month (*Hofmann et al., 2015*). Monitoring of the age at which each male sired its last litter showed that wild-type mice on average lost the ability to sire offspring around 16 months, while the *Ripk*3-knockout mice did not lose this ability until 22 months (*Figure 1I*).

To see if delay of aging phenotype was restricted to the reproductive system, we conducted histological analysis of major organs including small intestines, spleen, lung, liver, large intestines, kidney, heart, and brain of wild-type and *Ripk*3-knockout mice aged 8 weeks, 4 months, 18 months, and 24 months. We observed no differences between wild-type and age-matched *Ripk*3-knockout mice during the aging process (*Figure 1—figure supplement 6*).

However, when we examined the progenies sired by the aged *Ripk*3-knockout mice at a time wild-type mice had lost most of their reproductive activity, we found that they were less healthy no matter they were sired by wild-type or *Ripk*3-knockout mice than the progenies sired by young males, with higher rates of prenatal and postnatal death (*Figure 1—figure supplement 7A* and *Figure 1—source data 1*). A study into the possible reason for the unhealthy offspring revealed accumulated oxidative damage in the sperm DNA of aged *Ripk*3-knockout mice, measured as the level of 8-hydroxydeoxyguanosine (8-OHdG), a biomarker for the oxidative damage of DNA (*Chigurupati et al., 2008*; *Johnson et al., 2015*; *Paul et al., 2011*), was significantly higher in the sperm of 18-month-old wild type and *Ripk*3-knockout mice than in 4-month-old mice (*Figure 1—figure supplement 7B and C*).

The difference in whole body weight of 18-month-old wild-type and *Ripk*3-knockout mice was mainly due to the difference in their seminal vesicles and fat tissues (*Figure 1—figure supplement 8A–D*).

## RIPK3 expression in Spermatogonia, spermatocytes and Sertoli cells in testis

To investigate the underlining mechanism responsible for the delayed reproductive system aging phenotype, we first examined RIPK3 expression using immunohistochemistry methods (IHC). We noted that the cells inside wild-type seminiferous tubules were stained positively with anti-RIPK3 antibody (*Figure 2—figure supplement 1A*). In contrast, no staining was seen in the seminiferous tubules of *Ripk*3-knockout mice, confirming the specificity of the antibody (*Figure 2—figure supplement 1A*).

The specific cell types from testes were further analyzed by co-immunostaining of testis sections from sexually-mature wild-type mice (8 weeks) with antibodies against RIPK3 and other previously-described cell-type specific markers. RIPK3 expression was apparent in germ line spermatogonia expressing UTF1 (*Jung et al., 2014*; *van Bragt et al., 2008*) and in Sertoli cells expressing GATA-1

(*Tsai et al., 2006*). The testosterone-producing Leydig cells (marked by the HSD3B1(*Chang et al., 2011*) located outside of seminiferous tubules, however, did not express RIPK3 (*Figure 2A*). The RIPK3 expression in each of these cell types was further confirmed when testes were dissected and the cells were spread on a slide and analyzed again with co-immunostaining. The cell shapes changed due to spreading with this method, but the individual cells were more clearly visible. Consistent with the IHC staining results, spermatogonia and Sertoli cells were positive for RIPK3 staining while Leydig cells were not (*Figure 2B*). Moreover, the primary spermatocytes within seminiferous tubules that were not marked by IHC were now clearly visible when stained with the specific marker SMAD3 (*Hentrich et al., 2011*), and these cells expressed RIPK3 (*Figure 2B*). The fact that the cells within seminiferous tubules, the sperm-producing unit of testis, are all positive for RIPK3 expression raised a possibility that the age-associated depletion of these cells is through necroptosis.

## The RIPK3 substrate MLKL is phosphorylated in the seminiferous tubules of aged wild-type mice

Recall that RIPK3 transduces the necroptosis signal by phosphorylating the serine 345 of pesudokinase MLKL, we used an antibody against phospho-serine 345 of MLKL to analyze the testes of young and old mice. Phosphorylated MLKL (phospho-MLKL) was detected in seminiferous tubules in cells surrounding the center lumens in testes of 18-month-old wild-type mice, whereas no phospho-MLKL was detected in the same tissue area of 4-month-old wild-type mice, nor in 18-month-old *Ripk*3-knockout or *Mlkl*-knockout mice (*Figure 2C* and *Figure 2—figure supplement 1B*). A quantitative analysis of the phospho-MLKL staining of each age and genotype group is shown in *Figure 2D*. Consistently, phospho-MLKL was detected by western blotting in extracts from testes of 18- and 24-month-old wild-type mice but not in extracts from age-matched *Ripk*3-knockout mice (*Figure 2E*).

To further identify the exact cell type in the aged seminiferous tubules that show positive marker of necroptosis, we co-stained the testis sections with antibodies that specifically mark the different cell types in seminiferous tubules. As shown in *Figure 2F*, spermatogonia that specifically expressing UTF1 were co-stained with the anti-phospho-MLKL antibody. On the other hand, Sertoli cells did not show phospho-MLKL staining even though they do express RIPK3. Not surprisingly, Leydig cells that do not have RIPK3 expression also did not stain with the phosphor-MLKL antibody.

## Activation of apoptosis in Leydig cells during aging

The sex hormone-producing Leydig cells in testes do not express RIPK3, yet in old mice testis, the hormone level drops and Leydig cells are also gone. We therefore checked the cleavage status of procaspase-3 (a known marker of apoptosis) and procaspase-8 in the aged testes of wild-type and *Ripk*3-knockout mice using IHC. Cleaved procaspase-3 and cleaved procaspase-8 was detected in the wild-type Leydig cells of 18, and 36 month old mice, while no such signal was observed in age-matched *Ripk*3-knockout mice (*Figure 3A–D* and *Figure 3—figure supplement 1A and B*). The cleaved-caspase-3 was also detected by western blotting using extracts from the aged wild-type testes but not in *Ripk*3-knockout testes (*Figure 3—figure supplement 1C*). It is thus likely that Leydig cells undergo apoptosis, as a secondary response to necroptosis in seminiferous tubules during aging process.

## Caspase-8 levels decrease during aging in empty seminiferous tubules

We also used immunohistochemistry methods to examine the caspase-8 level in relative to RIPK3 in testes of wild-type mice of advanced age. In aged wild-type mice, caspase-8 levels decreased in the seminiferous tubules showing the sign of cell depletion (*Figure 3E and F*), and increased in the Leydig cells (*Figure 3—figure supplement 2*). This reduction in caspase-8 may explain how it is that necroptosis, but not apoptosis, occurs in the seminiferous tubules of aged mice.

## Knockout of *Mlkl* also delays the aging of mouse reproductive organs

The delayed testis aging phenotype of *Ripk*3-knockout mice and detection of necroptosis activation marker in spermatogonia in aged wild-type mice suggest that necroptosis might be part of the underlying cause of testis aging. To further investigate this possibility, we also characterized the aging-associated phenotype of *Mlkl*-knockout (*Mlkl*⁻/⁻) mice. We first weighed 15-month-old wild-type, *Ripk*3-knockout, and *Mlkl*-knockout mice. There was no significant difference between the

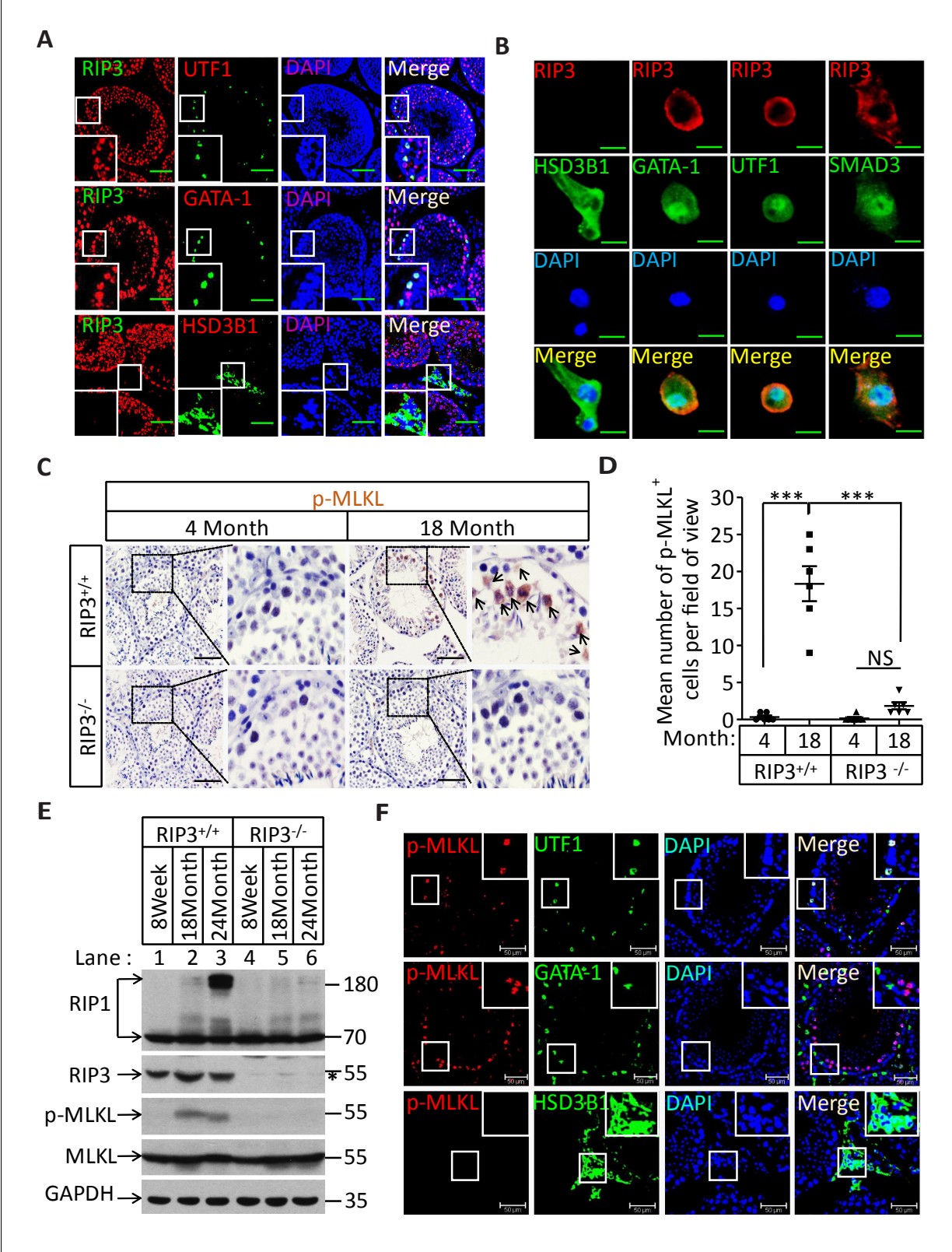

**Figure 2.** Necroptosis in seminiferous tubules of aged wild-type mice. (**A**) RIPK3 expression in spermatogonia, Sertoli cells, and spermatocytes. Immunofluorescence in an 8-week-old testis with antibodies against RIPK3 (red), HSD3B1 (Leydig cells specific protein, green), GATA-1 (Sertoli cells specific protein, green), and UTF1 (spermatogonium specific protein, green). Scale bar, 100 μm. (**B**) RIPK3 expression in germ line cells and Sertoli cells. Primary testis cells were isolated from wild-type testes, Immunofluorescence of Leydig cells, Sertoli cells, spermatogonia, and primary spermatocytes

*Figure 2 continued on next page*

*Figure 2 continued*

with antibodies against RIPK3 (red), HSD3B1 (green), GATA-1 (green), UTF1 (green), and SMAD3 (primary spermatocytes specific protein, green). Counterstaining with DAPI, blue. Scale bar, 10 μm. (C, D) Immunohistochemistry (IHC) of testes from *Ripk*3^+/+^ and *Ripk*3^-/-^ mice with phosphor-MLKL (p-MLKL) antibody. *Ripk*3^+/+^ (4 Months, n = 6; 18 Months, n = 6) and *Ripk*3^-/-^ (4 Months, n = 6; 18 Months, n = 6) mice were sacrificed and testes were harvested and stained with p-MLKL antibody in (C) (black arrows indicate cells with p-MLKL staining). p-MLKL^+^ cells were counted in five fields per testis and quantification in (D). Scale bar, 100 μm. Data represent the mean ± s.e.m. ***p<0.001. p values were determined with unpaired Student's *t*-tests. NS, not significant. (E) Western blot analysis of RIPK1, RIPK3, MLKL, and p-MLKL levels in the testis after perfusion, each group is representative of at least three mice. GAPDH was used as loading control. The asterisk (*) indicates non-specific bands. (F) Immunofluorescence in an 18-month-old testis with antibodies against p-MLKL (red, purple arrows indicate spermatogonium with p-MLKL staining), HSD3B1, GATA-1, and UTF1. Scale bar, 50 μm.

The following figure supplement is available for figure 2:

**Figure supplement 1.** RIPK3 expression in seminiferous tubules.

weights of *Mlkl*- and *Ripk*3-knockout mice, and mice of these knockout genotypes weighed less than wild-type mice at this age (*Figure 4A*). We also analyzed seminal vesicles and seminiferous tubules in aged *Mlkl*-knockout mice (15-month-old). Compared to the obvious aging that had occurred in wild-type mice, the seminal vesicles of *Mlkl*-knockout mice maintained a youthful appearance, exhibiting the same phenotype as *Ripk*3-knockout mice (*Figure 4B*). Furthermore, while the majority of seminal vesicles from 15-month-old wild-type mice weighed more than 1000 milligrams, almost all of the seminal vesicles from age-matched *Mlkl*- and *Ripk*3-knockout weighed less than 1000 milligrams (*Figure 4C*). Consistently, the testosterone levels of both *Mlkl*- and *Ripk*3-knockout mice were also significant higher than those of age-matched wild-type mice (*Figure 4D*). Further, very few (<2%) of the seminiferous tubules from *Mlkl*-knockout mice were empty at 15 months of age, similar to the tubules of *Ripk*3-knockout mice, while more than 12% of seminiferous tubules from the age-matched wild-type mice were already empty (*Figure 4E and F*). Finally, the fertility rates of both 16-month-old *Mlkl*- and *Ripk*3-knockout mice were also significant higher than those of age-matched wild-type mice (*Figure 4G*).

Although the wild-type, *Ripk*3-knockout, and *Mlkl*-knockout mice analyzed in this study were all C57BL/6 strain and were housed under the same condition, they were not littermates and their difference in aging should be interpreted with caution. We therefore used pharmacological means to further investigate the role of necroptosis in male reproductive system aging in the following experiments.

## Induction of necroptosis in testis depleted cells in seminiferous tubules

To directly demonstrate that necroptosis in testes is sufficient to cause the aging of the male reproductive system, we injected a combination of TNF-α, Smac mimetic, and caspase inhibitor z-VAD-FMK (henceforth 'TSZ')(*He et al., 2009*), a known necroptosis stimulus to the testes of 2-month-old mice. Injection of TSZ directly into the testis induced MLKL phosphorylation (*Figure 5A and B*). Phospho-MLKL was obviously present within the seminiferous tubules of TSZ-injected wild-type testes, but not TSZ-treated *Ripk*3-knockout or *Mlkl*-knockout testes, confirming the activation of necroptosis in testes following TSZ injection (*Figure 5C*). Moreover, when the cells were isolated from a wild-type testis and then treated with TSZ prior to staining with antibodies against phospho-MLKL and cell-type specific markers, cells in the seminiferous tubules, including spermatogonia, Sertoli cells, and spermatocytes, were stained positive for phospho-MLKL, whereas Leydig cells outside seminiferous tubules were negative (*Figure 5—figure supplement 1*). The consequences of necroptosis induction in testes became apparent 72 hr after a single TSZ injection. By this point, about 25% of wild-type seminiferous tubules were empty, whereas almost none of the seminiferous tubules from *Ripk*3- and *Mlkl*-knockout mice were affected (*Figure 5D and E*).

## Induction of necroptosis in testes accelerates aging of the male reproductive system

In addition to monitoring these short-term effects following a TSZ injection of 3-month-old mice, we waited for three additional months following the injection and assessed the long-term effects of induced necroptosis in mouse testes. Interestingly, three months after TSZ injection, the seminal

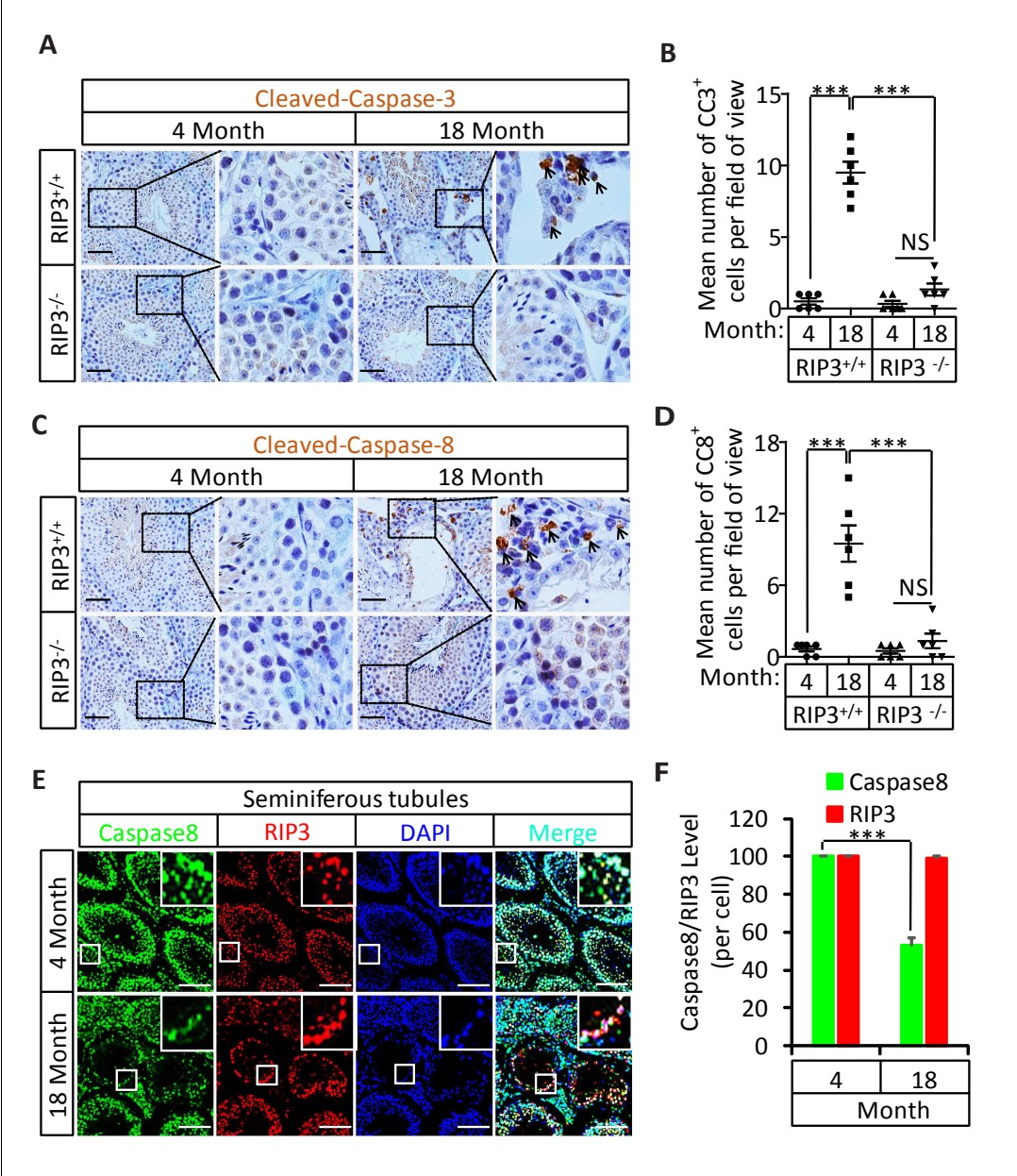

**Figure 3.** Activation of apoptosis in Leydig cells during aging. (**A, B**) IHC of testis from *Ripk+/+* and *Ripk3-/-* mice with Cleaved-Caspase-3 antibody. Mice were sacrificed, testes from *Ripk3+/+* (4 Month, n = 6; 18 Month, n = 6) and *Ripk3-/-* (4 Month, n = 6; 18 Month, n = 6) mice were harvested and stained with Cleaved-Caspase-3 antibody in (**A**) (black arrows for Leydig cells with Cleaved-Caspase-3 staining). Cleaved-Caspase-3+ cells were counted in six fields per testis and quantification in (**B**). Scale bar, 100 μm. Data represent the mean ± s.e.m. *p<0.05, ***p<0.001. p values were determined with unpaired Student's *t*-tests. (**C, D**) IHC of testis from *Ripk+/+* and *Ripk-/-* mice with Cleaved-Caspase-8 antibody. Mice were sacrificed, testes from *Ripk+/+* (4 Month, n = 6; 18 Month, n = 6) and *Ripk3-/-* (4 Month, n = 6; 18 Month, n = 6) mice were harvested and stained with Cleaved-caspase-8 antibody in (**C**) (black arrows for Leydig cells with Cleaved-caspase-8 staining). Cleaved-caspase-8+ cells were counted in six fields per testis and quantification in (**D**). Scale bar, 100 μm. Data represent the mean ± s.e.m. *p<0.05, ***p<0.001. p values were determined with unpaired Student's *t*-tests. (**E, F**) Caspase8 levels decrease during aging in empty seminiferous tubules. Immunofluorescence of testes from 4-month-old and 18-month-old wild-type mice with caspase8 and RIPK3 antibody in (**E**). The caspase8 levels were quantified in (**F**). Counterstaining with DAPI, blue. Scale bar, 100 μm.

The following figure supplements are available for figure 3:

**Figure supplement 1.** Activation of apoptosis in Leydig cells during aging.

**Figure supplement 2.** Caspase8 levels increase during aging in Leydig cells.

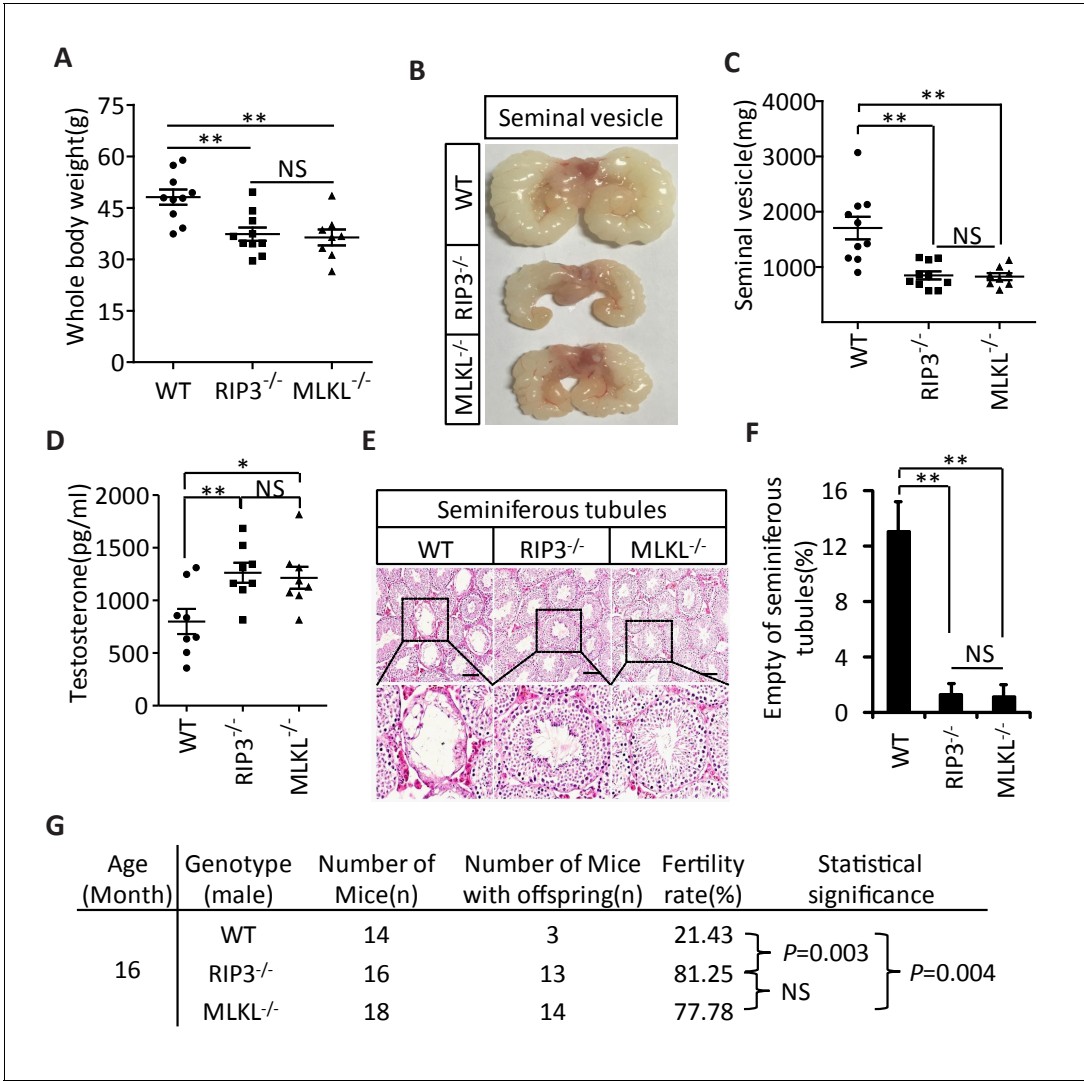

**Figure 4.** Aging of reproductive organs is delayed in *Mlkl*<sup>-/-</sup> mice. (A) Weight of WT (wild-type), *Ripk3*<sup>-/-</sup>, and *Mlkl*<sup>-/-</sup> male mice. WT (15 Month, n = 10), *Ripk3*<sup>-/-</sup>(15 Month, n = 10) and *Mlkl*<sup>-/-</sup> (15 Month, n = 8) male mice were weighed. Data represent the mean ± s.e.m. **p<0.01. p values were determined with unpaired Student's *t*-tests. NS, not significant. (B, C) Macroscopic features and weights of seminal vesicles. WT (15 month, n = 10), *Ripk3*<sup>-/-</sup> (15 Month, n = 10) and *Mlkl*<sup>-/-</sup>(15 Month, n = 8) male mice were sacrificed and the seminal vesicles were photographed and weighed. Data represent the mean ± s.e.m. **p<0.01. p values were determined with unpaired Student's *t*-tests. NS, not significant. (D) Serum testosterone levels of mice assayed using ELISA. Mice were sacrificed and the level of testosterone in serum from WT (15 Month, n = 8), *Ripk3*<sup>-/-</sup>(15 Month, n = 8) and *Mlkl*<sup>-/-</sup>(15 Month, n = 8) mice was measured using a testosterone ELISA kit. Data represent the mean ± s.e.m. *p<0.05, **p<0.01. p values were determined with unpaired Student's *t*-tests. NS, not significant. (E, F) H and E of testis sections from WT, *Ripk3*<sup>-/-</sup>, and *Mlkl*<sup>-/-</sup>mice. Testes from WT (15 Months, n = 8), *Ripk3*<sup>-/-</sup> (15 Months, n = 8), and *Mlkl*<sup>-/-</sup>(15 Month, n = 8) mice were harvested and stained with H and E in (E). The number of empty seminiferous tubules was counted based on H and E staining and quantification in (F), empty seminiferous tubules were counted in five fields per testis. Scale bar, 100 μm. Data represent the mean ± S.D. **p<0.01. p values were determined with unpaired Student's *t* tests. (G) Summary of the fertility rates of WT, *Ripk3*<sup>+/+</sup> and *Mlkl*<sup>-/-</sup> mice. One male mice of a given age was mated with a pairs of 10-week-old wild-type female mice for 3 months; females were replaced every 2 weeks. The number of male mice with reproductive capacity was counted (see Materials and methods). p values were determined using chi-square tests.

vesicles of wild-type recipient mice were as enlarged as those from mice older than 15 months. However, no such enlargement of seminal vesicles was observed in *Ripk3*- and *Mlkl*-knockout mice after the same TSZ treatment of their testes (*Figure 6A and B*). Additionally, more than 30% of the wild-type seminiferous tubules remained empty three months after the injection, while those of *Ripk3*- and *Mlkl*- knockout mice appeared completely normal without any observable loss of cells

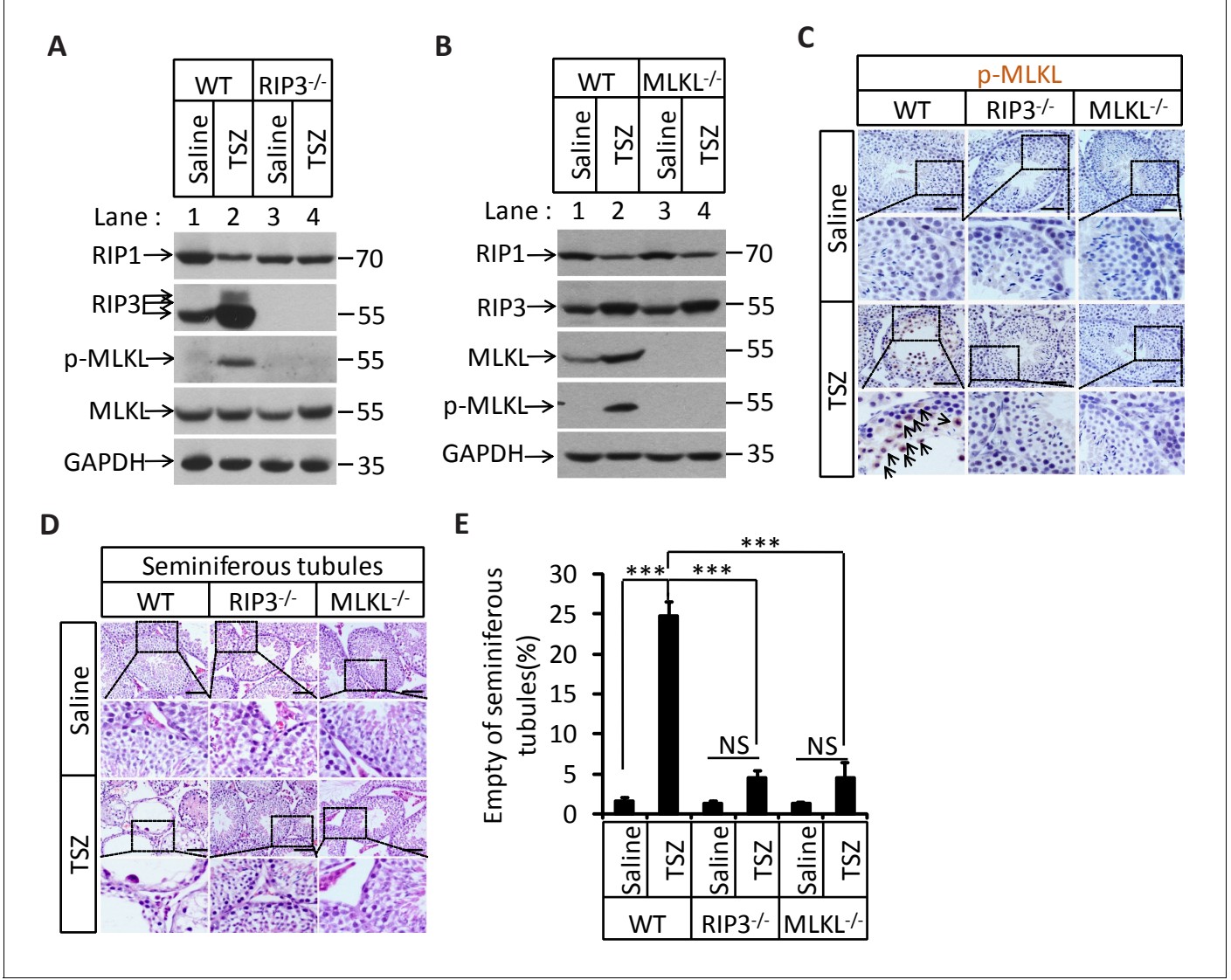

**Figure 5.** Induction of necroptosis in testis depleted cells in seminiferous tubules. (**A–E**) Testes of WT (2 Months, n = 6), *Ripk3*^-/- (2 Months, n = 6), and *Mlkl*^-/-(2 Month, n = 6) mice were injected with TSZ or saline (see Materials and methods); 72 hr after the injection, mice were sacrificed and the testes were harvested. The proteins were extracted from testes and were analyzed with western blotting in (**A, B**). GAPDH was used as a loading control. The testes were stained with p-MLKL antibody in (**C**) (black arrows indicate cells with p-MLKL staining). Scale bar, 100 μm. The testes were stained with H and E in (**D**). The number of empty seminiferous tubules was counted based on H and E staining and quantification in (**E**), empty seminiferous tubules were counted in five fields per testis. Scale bar, 100 μm. Data represent the mean ± S.D. ***p<0.001. p values were determined with unpaired Student's *t*-tests. NS, not significant.

The following figure supplement is available for figure 5:

**Figure supplement 1.** Activation of necroptosis in germ line stem cells and Sertoli cells in seminiferous tubules.

(*Figure 6C and D*). The lack of regeneration of cells in the seminiferous tubules indicated that necroptosis induction indeed has taken out their stem cells.

We also tested the fertility rate of TSZ-treated mice 3 month after the TSZ treatment. Control injection of saline into the testes of wild-type mice did not affect the fertility rate and the mice remained 100% fertile, but TSZ injection reduced the fertility rate by 87.5% (only 1 of 8 was fertile) (*Figure 6E*). In contrast, 6 out of 8 *Ripk3*-knockout mice and 7 out of 8 *Mlkl*-knockout mice were still fertile following TSZ injection (*Figure 6E*).

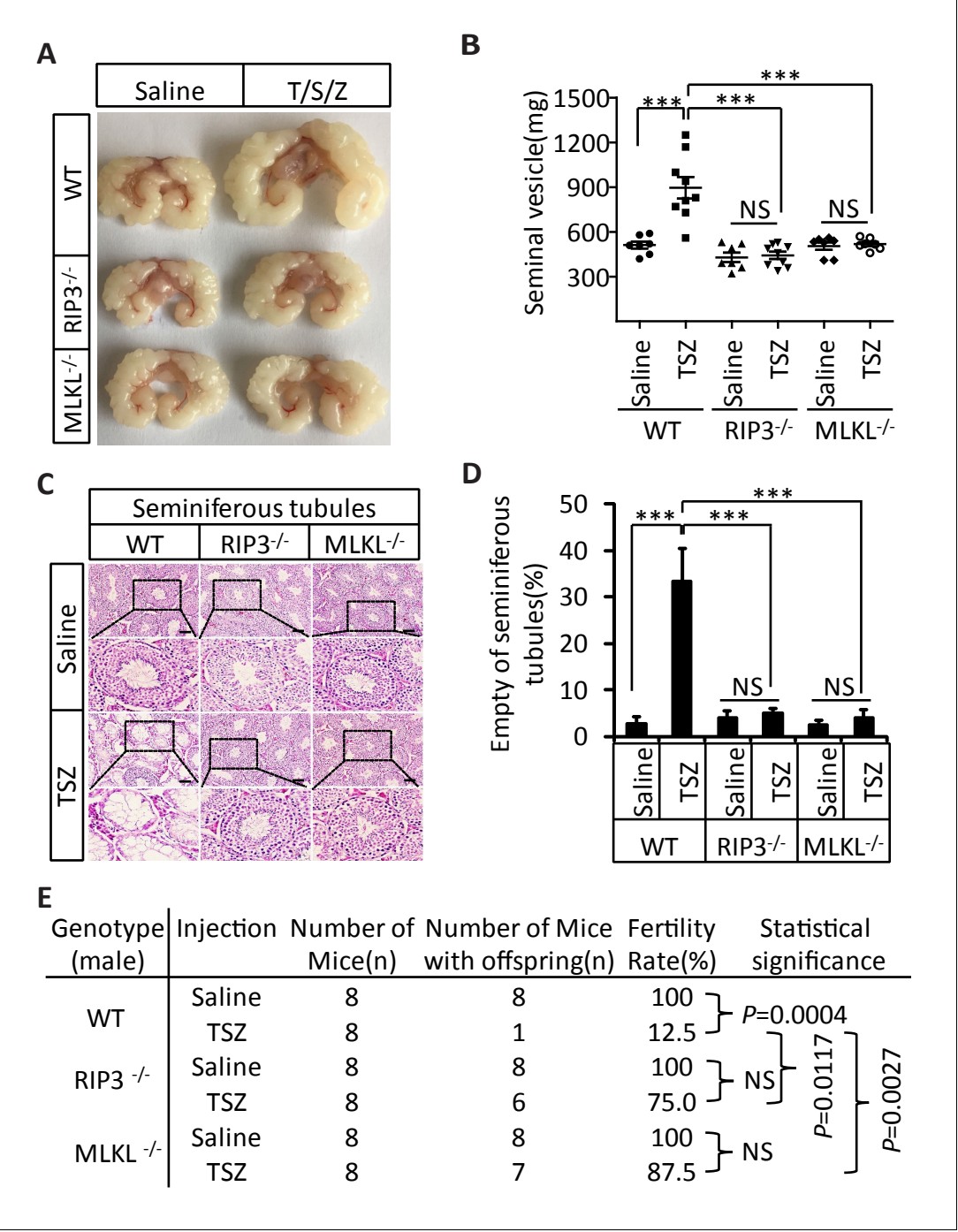

**Figure 6.** Induction of necroptosis in testes accelerates aging of the male reproductive system. (**A–D**) Testes from WT (3 Months, n = 9), *Ripk3*-/- (3 Months, n = 9), and *Mlkl*-/-(3 Month, n = 7) mice were injected with TSZ or saline (see Materials and methods) and were maintained in SPF facility for 3 months. Mice were then sacrificed and the seminal vesicles were photographed and weighed. Macroscopic features and weights of seminal vesicles form mice in (**A, B**). Data represent the mean ± s.e.m. ***p<0.001, p values were determined with unpaired Student's t-tests. Testes were harvested and stained with H and E in (**C**). The number of empty seminiferous tubules was counted based on H and E staining and quantification in (**D**). Empty seminiferous tubules were counted in five fields per testis. Scale bar, 100 μm. Data represent the mean ± S.D. ***p<0.001, p values were determined with unpaired Student's t-tests. NS, not significant. (**E**) Summary of the fertility rate of WT, *Ripk3*-/-, and *Mlkl*-/- male mice after injection with TSZ. Testes from WT (3 Months, n = 8), *Ripk3*-/- (3 Months, n = 8), and *Mlkl*-/-(3 Month, n = 8) male mice were injected with TSZ or saline (see Materials and methods) and mice were maintained in SPF

*Figure 6 continued on next page*

*Figure 6 continued*
for 3 months. One male mouse was mated with a pairs of 10-week-old female wild-type mice for 2 months; females were replaced every 2 weeks. The number of male mice with reproductive capacity was counted (see Materials and methods). p values were determined using chi-square tests.

## An RIPK1 kinase inhibitor blocks aging of the male reproductive system

The identification of the role of necroptosis in the aging of the mouse male reproductive system suggests the feasibility of a pharmaceutical intervention against the aging process. We therefore evaluated the effects of a newly-identified, highly-potent, and highly-specific RIPK1 kinase inhibitor from our laboratory (henceforth 'RIPA-56')(*Ren et al., 2017*) by incorporating it into mouse food at 150 mg/kg and 300 mg/kg doses. We first tested the effect of RIPA-56 on necroptosis in testes by injecting TSZ into testes of 2-month-old mice after feeding the mice with increasing concentrations of RIPA-56-containing chow for one week. RIPA-56 blocked the appearance of TSZ-induced phospho-MLKL in the testes in a dose-dependent manner, and was able to completely block necroptosis at the 300 mg/kg dose (*Figure 7A*, lane 4).

We subsequently chose the 300 mg/kg dose to continuously feed 13-month-old male wild-type mice for 5 months and analyzed these mice after 2 and 5 months to study the long-term effects of blocking necroptosis on testes. After two months, the mice fed with RIPA-56-containing diet weighed less than mice fed with control chow diet (*Figure 7B*). The seminal vesicles of the RIPA-56-treated mice retained the mass (mostly around 1000 milligrams), while the seminal vesicles from mice on normal chow grew significantly during the same period, with majority of them weighing more than 2000 milligrams (*Figure 7C and D*). Additionally, the testosterone level of RIPA-56-treated mice remained high, while that of control mice decreased (*Figure 7E*). Consistently, more than 12% of the seminiferous tubules of the control mice were empty, whereas hardly any seminiferous vesicles were empty in the RIPA-56-treated mice (*Figure 7F and G*). Finally, the fertility rates of the RIPA-56-treated mice were much higher than those of control mice with 19 out of 25 mice (76%) on the RIPA-56 diet were fertile while only 6 out of 23 mice (26%) on normal diet produced progeny (*Figure 7H*).

After 5 months, the difference in fertility was even more dramatic between mice on RIPA-56-containing and the ones on normal chow diet. At age of 18 months, 10 out of 15 mice (67%) on RIPA 56 diet still produced progenies while only 2 out of 15 mice (13%) on normal chow diet did (*Figure 7I*). Consistently, when we stained the testes from these mice with the anti-phospho-MLKL antibody, we observed abundant signals from the testes of mice on normal chow whereas hardly any phospho-MLKL signal was seen in mice on RIPA-56-containing diet (*Figure 7J and K*).

## Discussion

### Necroptosis promotes the aging phenotype of mouse male reproductive system

The above presented data indicated that the previously unknown physiological function of necroptosis is to promote the aging of male reproductive organs. We detected for the first time under physiological conditions the activation marker of necroptosis in spermatogonia of old testis. Consistently, mice with either of their core necroptosis execution components *Ripk3* and *Mlkl* deleted from their genome showed dramatic delay of male reproductive aging phenotype, both morphologically and functionally.

Interestingly, one dose of TSZ treatment applied locally to the young testes resulted in the male reproductive system aging phenotype, including the enlargement of seminal vesicles, the depletion of cells in the seminiferous tubules, and decreases in fertility rates in wild-type but not *Ripk3*-knockout or *Mlkl*-knockout mice, strongly suggests that necroptosis happening within testis is the cause of symptomatic male reproductive system aging.

However, the cellular and molecular events that lead to the eventual aging phenotype warrant extensive future studies. Although only in spermatogonia cells the phospho-MLKL signal was observed in aged testis, Sertoli cells in seminiferous tubules were also depleted. Sertoli cells do

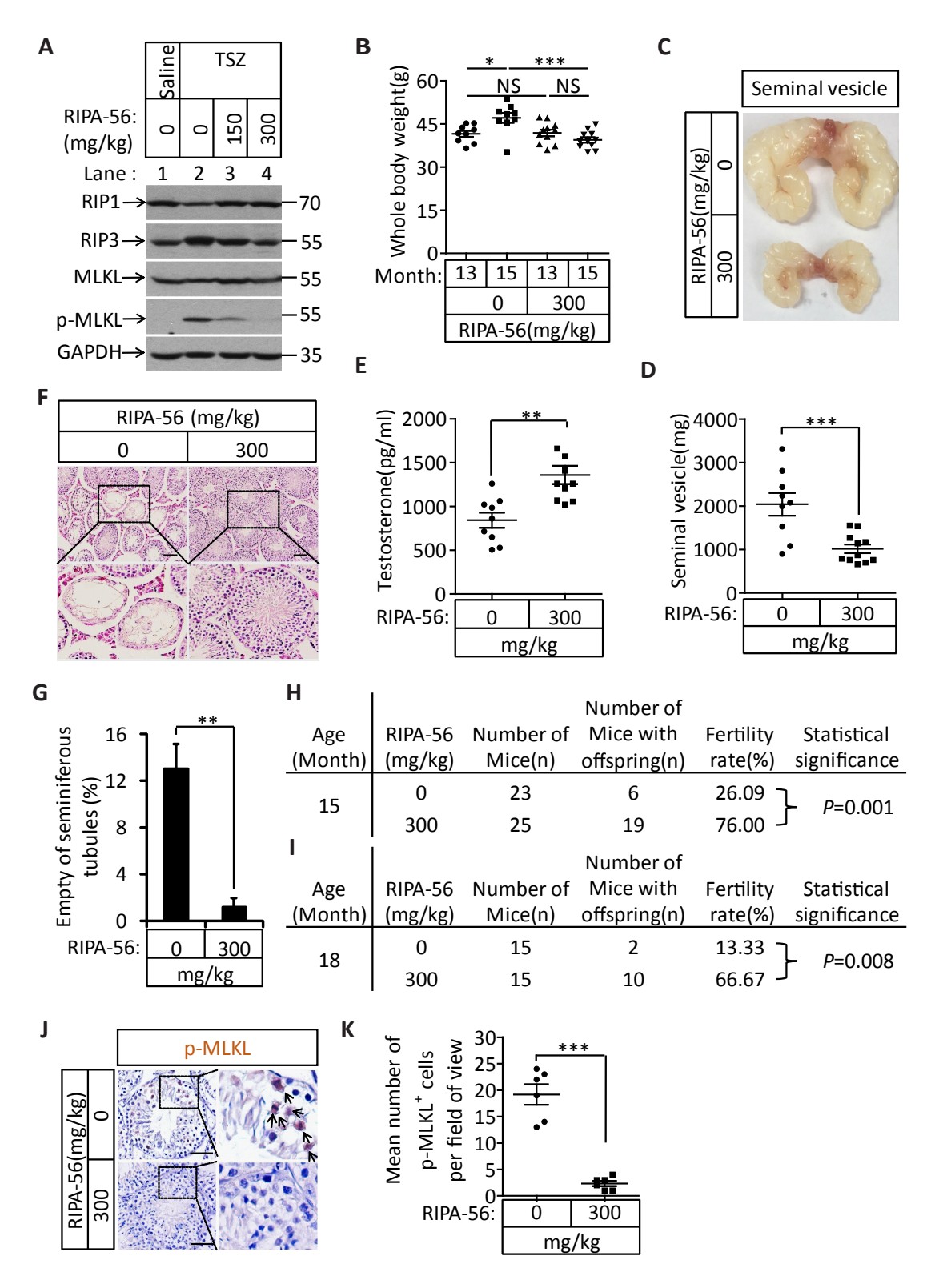

**Figure 7.** An RIPK1 inhibitor blocks aging of the male reproductive system. (**A**) Western blot analysis of RIPK1, RIPK3, MLKL, and p-MLKL levels in testes after injection with TSZ. The 2-month-old wild-type male mice continuously feed with RIPA-56 (0 mg/kg, n = 6; 150 mg/kg, n = 3; 300 mg/kg, n = 3) for one week. Testes were injected with TSZ (see Materials and methods); 72 hr after the injection, mice were sacrificed and the testes were harvested. The proteins were extracted from testes and were analyzed with western blotting. GAPDH was used as a loading control. (**B–H**) 13-month-old wild-type

*Figure 7 continued on next page*

*Figure 7 continued*

male mice were feed with AIN93G (RIPA-56:0 mg/kg) or AIN93G-RIPA-56 (RIPA-56:300 mg/kg) for 2 months in SPF facility. Mice were weighed before and after feed with RIPA-56 in (**B**). Data represent the mean ± s.e.m. *p<0.05, ***p<0.001. p values were determined with unpaired Student's *t*-tests. Mice were sacrificed and the seminal vesicles were photographed and weighed. Macroscopic features and weights of seminal vesicles form mice in (**C**, **D**). Data represent the mean ± s.e.m. ***p<0.001. p values were determined with unpaired Student's *t*-tests. Testosterone levels in serum from mice were measured using a testosterone ELISA kit in (**E**). Data represent the mean ± s.e.m. **p<0.01. p values were determined with unpaired Student's *t*-tests. The testes were harvested and stained with H and E in (**F**). The number of empty seminiferous tubules was counted based on H and E staining and quantification in (**G**). Empty seminiferous tubules were counted in five fields per testis. Scale bar, 100 µm. Data represent the mean ± S.D. **p<0.01. p values were determined with unpaired Student's *t*-tests. The fertility rate of each RIPA-56-treated (0 mg/kg, n = 23; 300 mg/kg, n = 25) male mice was assessed by mating it with four 10-week-old wild-type female mice successively (see Materials and methods). The number of mice with reproductive capacity was counted in (**H**). p values were determined using chi-square tests. (**I–K**) 13-month-old wild-type male mice were feed with AIN93G (RIPA-56:0 mg/kg) or AIN93G-RIPA-56 (RIPA-56:300 mg/kg) for 5 months in SPF facility. The fertility rate of each RIPA-56-treated (0 mg/kg, n = 15; 300 mg/kg, n = 15) male mice was assessed by mating it with four 10-week-old wild-type female mice successively (see Materials and methods). The number of mice with reproductive capacity was counted in (**I**). p values were determined using chi-square tests. After fertility test, mice were sacrificed and testes were harvested and stained with p-MLKL antibody in (**J**) (black arrows indicate cells with p-MLKL staining). p-MLKL[+] cells were counted in five fields per testis and quantification in (**K**). Scale bar, 100 µm. Data represent the mean ± s.e.m. ***p<0.001. P values were determined with unpaired Student's *t*-tests.

The following figure supplement is available for figure 7:

**Figure supplement 1.** TNF-α increased in the testes from aged wild-type mice.

express RIPK3 and undergo necroptosis in response to TSZ, however, whether these cells undergone necroptosis during the natural aging process and how their death related to necroptosis of spermatogonia are not known. Moreover, as testis age, the hormonal producing Leydig cells also lost. Leydig cells do not express RIPK3 thus unable to undergo necroptosis. Interestingly, in *Ripk*3- and *Mlkl*-knockout mice, and in mice fed with the RIPK1 inhibitor, Leydig cells are spared. The death of Leydig cells, mostly like through apoptosis since active caspase-8 and caspase-3 were seen on these cells in aging testis, is obviously subsequential to necroptotic death somewhere else. We have no idea how and what signal Leydig cells receive from necroptotic cells to activate apoptosis of their own. In addition to the depletion of spermatogonium stem cells, Sertoli cells, and Leydig cells during testis aging, the most striking feature of aging mouse male sex organ is the enlargement of seminal vesicles. Such an enlargement clearly involves growth of epithelial cells of this auxiliary organ. What growth signal the epithelial cells of seminal vesicle receive in response to the depletion of cells in testis is also enigmatic.

## Necroptosis-promoted male reproductive system aging offers an evolutionary advantage at species level

The fact that knocking out either of the core necroptosis executing component from mouse genome or feeding mice with a chemical necroptosis inhibitor results in prolonged male reproductive longevity indicates that necroptosis actively promotes male reproductive system aging in wild-type mice. However, the progeny produced by aged mice with artificially extended reproductive longevity were not healthy. The cause for these unhealthy pups may be multiple, including age-related accumulation of DNA damages in the old sperm and other organs. Therefore, although mice without the core components of the necroptosis pathway maintain their reproductive activity into advanced ages (well beyond the age when wild-type mice have largely lost such capacity), these age-associated changes still caused deleterious effects on their progeny.

We therefore propose that necroptosis in testis is a physiological response to yet-to-be-identified, age-related, TNF family of cytokine(s) that transduces necroptosis signal through the canonical RIPK1-RIPK3-MLKL pathway. The necroptotic death of cells in testis then triggers the other downstream age-related phenotypes such as enlargement of seminal vesicles, decreased testosterone levels and weight gain. Indeed, we observed an elevated TNF-α level in the old testis of wild-type mice (*Figure 7—figure supplement 1*). Interestingly, no such elevation was seen in the testis of age-matched *Ripk*3-knockout mice, indicating that such an elevation may be augmented through a feed forward mechanism. Whether TNF-α or other members of the TNF family of cytokines truly contributes to testis aging is not known and should be an interesting research topic for the future studies.

Necroptosis-instigated reproductive system aging effectively eliminates old animals from the reproductive pool. Given that aged animals carry significantly more DNA damage than younger animals, their elimination from the mating pool results in healthier pups overall, an outcome that would confer an evolutionary advantage over (a population) of animals that do not thusly employ a necroptosis program in their testes.

Interestingly, when wild-type mice were fed with food containing an RIPK1 inhibitor prior to the onset of reproductive system aging (13 months), the aging of the male reproductive system could be blocked and the effect lasted for 5 months when the experiment was terminated. This finding not only further confirms that necroptosis is the mechanism underlying male reproductive system aging, but also demonstrates an apparently-effective way to delay it.

## Materials and methods

### Mice

The $Ripk3^{-/-}$ ($Ripk3$-knockout) mice (C57BL/6 strain) were generated as described previously (*Newton et al., 2004*). The $Mlkl^{-/-}$ ($Mlkl$-knockout) mice were generated by co-microinjection of in vitro-translated Cas9 mRNA and gRNA into the C57BL/6 zygotes. Founders were screened with T7E1 assays and were validated by DNA sequencing. Founders were intercrossed to generate bi-allelic $Mlkl^{-/-}$ mice. The gRNA sequence used to generate the knockout mice was GTAGCAG TTGCAAATTAGCGTGG. C57BL/6 wild-type (WT, $Ripk3^{+/+}$) mice were obtained from Vital River Laboratory Co (Beijing, China). WT, $Ripk3^{-/-}$, and $Mlkl^{-/-}$ mice were produced and maintained at the SPF animal facility of the National Institute of Biological Sciences, Beijing. Animals for the aging study were produced by mating wild-type males with wild-type females purchased from Vital River Laboratory Co; $Ripk3^{-/-}$ mice were produced by mating $Ripk3^{-/-}$ males with Ripk3$^{-/-}$ females; $Mlkl^{-/-}$ mice were produced by mating $Mlkl^{-/-}$ males with $Mlkl^{-/-}$ females. Animals (male) for the aging study were housed under the same conditions after birth. Animals used for the hormone and the fertility tests were housed individually in an SPF barrier facility.

### Antibodies and reagents

The Antibody against RIPK3 (#2283; WB, 1:1000; IHC, 1:100) were obtained from ProSci (San Diego, CA). Other antibodies used in this study were: anti-GAPDH-HRP (M171-1, MBL (Nagoya, Jpan), 1:5000), anti-RIPK1 (#3493S, Cell Signaling (Danvers, MA), 1:2000), anti-MLKL (AO14272B, ABGENT (San Diego, CA), 1:1000), anti-Mouse-phospho-MLKL (ab196436; WB, 1:1000; IHC, 1:100), anti-GATA-1 (sc-265, IHC, 1:200), anti-cleaved-caspase-3 (#9661, Cell Signaling; WB, 1:1000; IHC 1:100), anti-caspase-8 (proteintech, 66093–1-Ig, 1:100), anti-cleaved-caspase-8 (8592T, Cell Signaling; IHC 1:100), anti-HSD3B1 (ab150384, 1:200) (Abcam, Cambridge Great Britain), anti-SMAD3 (MA5-15663, Thermo (Waltham, MA), 1:200), 8-OHdG (N45.1; Genox (Baltimore, MD); 1:200), and anti-UTF1 (#MAB4337, EMD Millipore (Billerica, MA), 1:200). DPBS (Dulbecco's Phosphate-Buffered Saline) was purchased from Thermo. Lectin from *Datura stramonium*, Sodium L-lactate, Deoxyribonuclease I from bovine pancreas, and MEM Non-essential Amino Acid Solution (100×) were purchased from Sigma. RIPA-56 (RIPK1 inhibitor) was generated as described in *Ren et al. (2017)*.

### Cell cultures

Primary testis cells were cultured in DMEM:F12 Medium (Hyclone) supplemented with 10% FBS (Invitrogen, Carlsbad, CA) and penicillin/streptomycin (Invitrogen).

### Harvesting of tissues

Mice were euthanized using avertin (20 mg ml$^{-1}$). Animals were euthanized one by one immediately before dissection, and the dissection was performed as rapidly as possible by a team of several trained staff members working in concert on a single mouse. Major organs were removed, cut into appropriately-sized pieces, and either flash-frozen in liquid nitrogen and stored at −80°C or placed in formalin (Using Bouin's fixative for testis) for preservation. After several days of formalin fixation at room temperature, tissue fragments were transferred to 70% ethanol and stored at 4°C. Blood was collected by cardiac puncture, and was allowed to coagulate for the preparation of serum.

## Western blotting

Western blotting was performed as previously described (*Wang et al., 2014*). Briefly, cell pellet samples were collected and re-suspended in lysis buffer (100 mM Tris-HCl, pH 7.4, 100 mM NaCl, 10% glycerol, 1% Triton X-100, 2 mM EDTA, Roche complete protease inhibitor set, and Sigma phosphatase inhibitor set), incubated on ice for 30 min, and centrifuged at 20,000 × g for 30 min. The supernatants were collected for western blotting. Testes tissue samples were ground and re-suspended in lysis buffer, homogenized for 30 s with a Paddle Blender (Prima, PB100), incubated on ice for 30 min, and centrifuged at 20,000 × g for 30 min. The supernatants were collected for western blotting.

## ELISA

Mice (male) used for the hormone tests were housed individually in an SPF barrier facility. Mice were sacrificed and blood was clotted for two hours at room temperature before centrifugation at approximately 1,000 × g for 20 min. Mice blood sera was collected and assayed immediately or was stored as sample aliquots at −20℃. The testosterone/FSH/LH levels were measured with ELISA kits (BIO-MATIK, EKU07605, EKU04284, EKU05693); the SHDG level was measured with an ELISA kit (INSTRUCTION MANUAL, SEA396Mu). The ELISA assays were performed according to the manufacturer's instructions.

## Histology, immunohistochemisitry, and immunofluorescence

Paraffin-embedded specimens were sectioned to a 5 μm thickness and were then deparaffinized, rehydrated, and stained with haematoxylin and eosin (H and E) using standard protocols. For preparation of immunohistochemistry samples, sections were dewaxed, incubated in boiling citrate buffer solution for 15 min in plastic dishes, and subsequently allowed to cool down to room temperature over 3 hr. Endogenous peroxidase activity was blocked by immersing the slides in hydrogen peroxide buffer (10%, Sinopharm Chemical Reagent) for 15 min at room temperature and were then washed with PBS. Blocking buffer (1% bovine serum albumin in PBS) was added and the slides were incubated for 2 hr at room temperature. Primary antibody against phospho-MLKL, cleaved-caspase-3, and 8-OHdG was incubated overnight at 4℃ in PBS. After three washes with PBS, slides were incubated with secondary antibody (polymer-horseradish-peroxidase-labeled anti-rabbit, Sigma) in PBS. After a further three washes, slides were analyzed using a diaminobutyric acid substrate kit (Thermo). Slides were counter stained with haematoxylin and mounted in neutral balsam medium (Sinopharm Chemical).

Immunohistochemistry analysis for RIPK3 was performed by incubating the tissue slides with the indicated antibodies overnight at 4℃ in PBS. After three washes with PBS, slides were incubated with DyLight-561 conjugated goat anti-rabbit secondary antibodies (Life) in PBS for 8 hr at 4℃. After additional three washes, slides were incubated with HSD3B1, GATA-1, or UTF1 antibody overnight at 4℃ in PBS. The slides were then washed three more times before incubated with DyLight-488 conjugated goat anti-mouse/rat secondary antibodies (Life) for 2 hr at room temperature in PBS. The slides were then washed in PBS, and cell nuclei were then counter-stained with DAPI (Invitrogen) in PBS. The fluorescence images were observed using a Nikon A1-R confocal microscope.

## TSZ injection of testis

WT, *Ripk3*[-/-], and *Mlkl*[-/-] mice were anaesthetized with injection of 20 μl g$^{-1}$ avertin (20 mg ml$^{-1}$). The abdomen was opened with surgical scissors and the testes were taken out one-by-one. Each testis was then injected with 20 ng ml$^{-1}$ TNF-α in the presence of Smac mimetic (100 nM) and z-VAD-fmk (10 μM) in 20 μl. Each testis was then put back into the cavity, and the abdomen was sutured (*Hooley et al., 2009*). Mice were maintained in an SPF animal facility.

## Sperm count

Mice were sacrificed and their epididymides were harvested. Each epididymis was punctured with a 25-gauge needle. Sperm were extruded with tweezers from the epididymis and collected in 2 ml of PBS. The solution was strained with a cell strainer (40 μm, BD Falcon), and 5 μl was taken out and diluted in 95 μl PBS. The number of sperm was counted using a cell-counting chamber under a microscope (*Schürmann et al., 2002*).

## Mating and fertility test

Mice (male) used for the fertility tests were housed individually in an SPF barrier facility. To assess vaginal patency, mice were examined daily from weaning until vaginal opening was observed. The fertility rate of males was determined via a standard method (*Cooke and Saunders, 2002*; *Hofmann et al., 2015*) by mating a male with a series of pairs of 10-week-old wild-type females for 3 months; females were replaced every 2 weeks (females were either from our colony or purchased from Vital River Laboratory Co(C57BL/6)). Each litters was assessed from the date of the birth of pups; when pups were born but did not survive, we counted and recorded the number dead pups; for females that did not produce offspring, the number of pups was recorded as '0'(did not produce a litter with a proven breeder male for a period of 2 months). The number of male mice with reproductive capacity was recorded.

The reproductive longevity of males was determined by continuously housing 2-month-old $Ripk3^{+/+}$ and $Ripk3^{-/-}$ males with a pairs of 10-week-old wild-type females, the females being replaced every 2 months, until males ceased reproducing (calculated as the age at birth of the litter less 21 days) (*Hofmann et al., 2015*).

## Isolation of cells from testes

Testes from 8 week old mice were collected using a previously-reported protocol (*Chang et al., 2011*). Briefly, a testis was placed in Enriched DMEM:F12 (Hyclone) media and placed on ice. After removal of the tunica albuginea of a testis, the seminiferous tubules were dissociated and transferred immediately into 10 mL of protocol enzymatic solution 1. Tubules were incubated for 15–20 min at 35°C in a shaking water bath at 80 oscillations (osc)/min and were then layered over 40 mL 5% Percoll/95% 1 × Hank's balanced salt solution in a 50 mL conical tube and allowed to settle for 20 min. Leydig cells were isolated from the top 35 mL of the total volume of Percoll. The bottom 5 mL of Percoll was transferred to a fresh 50 mL conical tube containing 10 mL enzymatic solution 2. Tubules were incubated for 20 min at 35°C and 80 osc/min. After incubation, 3 mL charcoal-stripped FBS was immediately added to halt the digestion. All fractions were mixed and immediately centrifuged at 500 × g at 4°C for 10 min. Pellets were re-suspended in PBS and washed three times, then cultured in DMEM:F12(10%FBS) medium at 37°C.

## RIPA-56 feeding experiment

RIPA-56 in the AIN93G (LAD3001G) at 150 or 300 mg/kg was produced based on the Trophic Animal Feed High-tech Co's protocol. Cohorts of 13-month-old wild-type male mice were fed with AIN93G or AIN93G containing RIPA-56 (RIPA-56:300 mg/kg) for 2 months in an SPF facility; each male mouse was then mated with four 10-week-old wild-type female mice successively. The number of male mice with reproductive capacity were recorded.

## Statistical analysis

All experiments were repeated at least twice. Data represent biological replicates. Statistical tests were used for every type of analysis. The data meet the assumptions of the statistical tests described for each figure. Results are expressed as the mean ± s.e.m or S.D. Differences between experimental groups were assessed for significance using a two-tailed unpaired Student's t-tests implanted in GraphPad prism5 or Excel software. Fertility rates were assessed for significance using chi-square tests (unpaired, two-tailed) implemented in GraphPad prism5 software. The *p<0.05, **p<0.01, and ***p<0.001 levels were considered significant. NS, not significant.

## Acknowledgements

We thank Drs. Mengqiu Dong, Joseph Goldstein and Michael Brown for critically reading the manuscript and thank Dr. John Snyder and Mr. Alex Wang for editing the manuscript. This work was supported by a National Basic Science 973 grants (2013CB530805) from the Chinese Ministry of Science and Technology and by the Beijing Municipal Commission of Science and Technology. The funders had no role in study design, data collection and interpretation, or the decision to submit the work for publication.

# Additional information

## Funding

| Funder | Author |
| --- | --- |
| National Basic Science 973 grant | Xiaodong Wang |

The funders had no role in study design, data collection and interpretation, or the decision to submit the work for publication.

## Author contributions

DL, Conceptualization, Resources, Data curation, Formal analysis, Supervision, Funding acquisition, Writing—original draft, Project administration, Writing—review and editing; LM, Conceptualization, Data curation, Validation, Investigation, Visualization, Methodology, Writing—original draft, Project administration, Writing—review and editing; TX, Conceptualization, Data curation, Investigation; YS, Data curation, Formal analysis, Investigation, Methodology; XL, ZZ, Resources, Methodology; XW, Resources, Supervision, Methodology

## Author ORCIDs

Dianrong Li, http://orcid.org/0000-0002-5564-3033
Xiaodong Wang, http://orcid.org/0000-0001-9885-356X

## Ethics

Animal experimentation: Animal experimentation: Animal care and use followed the institutional guidelines of the National Institute of Biological Sciences (NIBS), Beijing (Approval ID: NIB-SLuoM15C) and the Regulations for the Administration of Affairs Concerning Experimental Animals of China.

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
