## [Decision Letter]

Thank you for submitting your article "Necroptosis promotes the Aging of the Male Reproductive System in Mice" for consideration by *eLife*. Your article has been favorably evaluated by Sean Morrison (Senior Editor) and four reviewers, one of whom is a member of our Board of Reviewing Editors. The following individual involved in review of your submission has agreed to reveal his identity: Manolis Pasparakis (Reviewer #2).

The reviewers have discussed the reviews with one another and the Reviewing Editor has drafted this decision to help you prepare a revised submission.

Summary:

This is an interesting study that describes a delayed aging phenotype in the male reproductive tissue of MLKL and RIPK3 knockout mice. These studies are complemented by gain-of-function and loss-of-function pharmacological studies. The reported findings are important as they identify male reproductive aging as a physiological process regulated by necroptosis. Although the implication of these findings for human reproductive aging remains to be assessed, the mouse data presented here and particularly the studies with the RIPK1 inhibitor may provide a possible therapeutic perspective.

Essential revisions:

Overall, the presented data are convincing and support the authors' main conclusions. However, a number of questions concerning the data shown require clarification.

1) The nature of the aging phenotype is unclear because it involves different cell types that appear to show different behaviors. For example, the decline in testosterone with age (blocked by RIPK3 and MLKL KO) is presumably due to Leydig cells that do not express RIPK3 (subsection “Activation of apoptosis in Leydig cells during aging”) and are proposed to die by apoptosis secondarily to a primary necroptosis response by other cells, possibly including spermatogonia and sertoli cells. However, the seminal vesicle response is different – is this a proliferation response or a cell death response – and what is the proximal cause? All of these phenotypes may result from a single defect; indeed, the authors speculate that death receptors may be responsible for triggering spermatogonial necroptosis and reproductive aging. However, this potential role of germ cell necroptosis represents only one possible explanation of these aging phenotypes that may involve specific defects in more than one cell type. The use of whole body knockout mice (rather than conditional KO mice) and pharmacological approaches in the study limits the ability to deduce mechanism. It is therefore important that alternative explanations of the observed phenotype are discussed.

2) The decrease in sperm number observed in age wild type mice may not completely explain the reduced reproductive longevity shown in Figure 1. In the absence of studies of mature sperm, in vitro fertilization experiments, or other more in-depth implantation analyses, the authors should be careful with this conclusion and discuss these data in more detail. The conclusion that all observed differences have a single cause might not be correct.

3) The authors speculate that death receptors may be responsible for triggering spermatogonial necroptosis and reproductive aging. Testing the functional role of the different death receptors in this process is beyond the scope of this manuscript, assessment of the expression of death ligands and receptors in the testes of young and aged mice could provide evidence strengthening this assumption. Moreover, assessment of markers of inflammation in these tissues (e.g. immune cell infiltration, cytokine/chemokine expression) would also provide helpful data on the possible role of an immune response in male reproductive organ aging.

4) The phospho-MLKL stainings support a role of necroptosis, but there is no evidence in the paper that cells in the seminiferous tubules actually die. In light of recent work by the Green lab showing that phosphorylated MLKL is not an irreversible marker of cell death, it would be helpful if the authors could assess cell death directly, for example by TUNEL or TEM in the tissues stained with the p-MLKL antibodies.

---

## [Author Response]

*Essential revisions:*

*Overall, the presented data are convincing and support the authors' main conclusions. However, a number of questions concerning the data shown require clarification.*

*1) The nature of the aging phenotype is unclear because it involves different cell types that appear to show different behaviors. For example, the decline in testosterone with age (blocked by RIPK3 and MLKL KO) is presumably due to Leydig cells that do not express RIPK3 (subsection “Activation of apoptosis in Leydig cells during aging”) and are proposed to die by apoptosis secondarily to a primary necroptosis response by other cells, possibly including spermatogonia and sertoli cells. However, the seminal vesicle response is different – is this a proliferation response or a cell death response – and what is the proximal cause? All of these phenotypes may result from a single defect; indeed, the authors speculate that death receptors may be responsible for triggering spermatogonial necroptosis and reproductive aging. However, this potential role of germ cell necroptosis represents only one possible explanation of these aging phenotypes that may involve specific defects in more than one cell type. The use of whole body knockout mice (rather than conditional KO mice) and pharmacological approaches in the study limits the ability to deduce mechanism. It is therefore important that alternative explanations of the observed phenotype are discussed.*

The enlargement of seminar vesicles when mice age is a well-known phenomenon and lack of such change in RIP3 and MLKL knockout mice was the first visual clue to us that necroptosis might have a role in mouse reproductive organ aging. Our interpretation that the observed enlargement of seminal vesicles is a consequence of necroptosis in the testis is mainly based on the experimental result showing that a single locally injected necroptosis-inducing agent (henceforth ‘TSZ’) into testes resulted in enlarged seminal vesicles in wild type mice while RIP3 and MLKL knockout mice received the same injection did not show such a phenotype. We agree with the reviewers’ comments that since testis consists of several cell types with different cell death responses and our most straight forward interpretation that spermtogonia stem cells’ necroptosis is the trigger for male reproductive system aging may be a bit simplistic. We thus change the text to “necroptosis is part of the process underling the mouse male reproductive organ aging” and added a paragraph in the Discussion to entertain more complicated explanations.

*2) The decrease in sperm number observed in age wild type mice may not completely explain the reduced reproductive longevity shown in Figure 1. In the absence of studies of mature sperm,* in vitro *fertilization experiments, or other more in-depth implantation analyses, the authors should be careful with this conclusion and discuss these data in more detail. The conclusion that all observed differences have a single cause might not be correct.*

We agree with the reviewers’ comments that the reduced reproductive longevity in wildtype mice shown in Figure 1 may result from a combinational effect of several factors in addition to decrease in sperm counts. These include testosterone level drop, empty of seminiferous tubules, drop of pituitary hormone levels, and enlargement of seminar vesicles. We incorporated these factors in the conclusion in our new manuscript.

*3) The authors speculate that death receptors may be responsible for triggering spermatogonial necroptosis and reproductive aging. Testing the functional role of the different death receptors in this process is beyond the scope of this manuscript, assessment of the expression of death ligands and receptors in the testes of young and aged mice could provide evidence strengthening this assumption. Moreover, assessment of markers of inflammation in these tissues (e.g. immune cell infiltration, cytokine/chemokine expression) would also provide helpful data on the possible role of an immune response in male reproductive organ aging.*

Our speculation that death receptors participate in testis aging is mainly based the fact that RIP1 kinase inhibitor could blocks such aging phenotype and RIP1 is the only known signaling molecule that transduces necroptosis signal from death receptors to RIP3/MLKL. The suggestion to check the level of death receptors in aging testis is a good one and we added a western blotting analysis showing that TNFα level was increased in the aged wild type testes but not in the aged RIP3-knockout mice (new Figure 7—figure supplement 1). This data also indicated that the increase of this cytokine may be regulated in a feedforward fashion. We did not go further in this manuscript on the details of inflammatory elements in testis aging for a simple fact that interpretations of these observations are complicated and not helpful to the main conclusion of this manuscript.

*4) The phospho-MLKL stainings support a role of necroptosis, but there is no evidence in the paper that cells in the seminiferous tubules actually die. In light of recent work by the Green lab showing that phosphorylated MLKL is not an irreversible marker of cell death, it would be helpful if the authors could assess cell death directly, for example by TUNEL or TEM in the tissues stained with the p-MLKL antibodies.*

Although phospho-MLKL may not be an irreversible necroptotic marker, it is still the best characterized, necroptosis specific activation marker. To further strengthen this point, we added one more experimental result (new Figure 7) showing that testes of wild type mice on RIP1 inhibitor-containing diet for 5 months (now 18-month old) were devoid of phospho-MLKL signal whereas that from the control group on normal chow showed abundant staining. The non-irreversible nature of phospho-MLKL as described by Green’s group may contribute to the persistence of the signal so we could robustly detect them in the aging testes. On the other hand, TUNEL or TEM assay is not specific for necroptosis. We therefore did not use them.